# Screening of Plant Pollen Sources, Polyphenolic Compounds, Fatty Acids and Antioxidant/Antimicrobial Activity from Bee Pollen

**DOI:** 10.3390/molecules27010117

**Published:** 2021-12-26

**Authors:** Yusuf Can Gercek, Saffet Celik, Sinan Bayram

**Affiliations:** 1Department of Biology, Faculty of Science, Istanbul University, 34134 Istanbul, Turkey; 2Centre for Plant and Herbal Products Research-Development, 34134 Istanbul, Turkey; 3Technology Research and Development Application and Research Center, Trakya University, 22030 Edirne, Turkey; saffetcelik@trakya.edu.tr; 4Department of Medical Services and Techniques, Vocational School of Health Services, Bayburt University, 69000 Bayburt, Turkey

**Keywords:** bee pollen, fatty acid, polyphenolic profile, in vitro antibacterial activity

## Abstract

In this study, the botanical origin, total flavonoid and phenolic content, antioxidant activity, phenolic profile and fatty acid composition of mixed bee pollen loads collected in Bayburt, Turkey, were determined. In addition to these assays, antibacterial activity of bee-collected pollen extract (BCPE) against a variety of food-borne pathogenic bacteria was determined in vitro. Pollen loads were classified into five botanical families based on their color: Asteraceae, Fabaceae, Campanulaceae, Cistaceae and Rosaceae. Total flavonoid, total phenolic, CUPRAC and CERAC concentrations were 173.52 mg GAE/g, 79.21 mg QE/g, 85.59 mg Trolox/g and 118.13 mg Trolox/g, respectively. Twenty-three phenolic compounds were scanned in bee pollen extract by LC-MS/MS, with rutin being the most abundant. Cis-4,7,10,13,16,19 docosahexaenoic acid was the predominant fatty acid, followed by cis-11-eicosenoic acid, palmitic acid, and alfa linolenic acid. In addition, the agar well diffusion (AWD) and micro-broth dilution methods were used to determine of the antibacterial activity of the BCPE sample. MIC values were observed to vary between 2.5–5 mg/mL for Gram-positive bacteria and 5–10 mg/mL for Gram-negative bacteria. These findings indicate that bee pollen could be a potential source of antioxidants and antimicrobials.

## 1. Introduction

In recent years, as the global population has increased, the global demand for nutrition has increased as well. People, particularly in developed and developing countries, prefer to eat healthily, and changes in consumer dietary habits occur [1]. With the understanding that natural/functional foods have a positive effect in the prevention and treatment of diseases, these foods have become more popular among the public [2]. Accordingly, numerous natural products, including bee products, have established a presence on market shelves and have begun to garner consumer interest. The bee products include honey, bee pollen, bee bread, royal jelly, beeswax, bee venom, apilarnil and beeswax [3]. One of these products, bee pollen, is collected from plant flowers by the honey bee. The collected flower pollen is accumulated as corbicular pellets in the pollen baskets located on the hind legs of the honey bee, and bee pollen is formed [4]. Bee pollen is classified in two groups according to its flower source: monofloral (the major taxon needs to be not less than 80%) and polyfloral (which contains pollen from more than one plant taxon) [4].

Pollen is the primary source of protein for bees and thus can be considered a potential source of energy and protein for humans [1]. Pollen contains essential amino acids such as threonine, phenylalanine, histidine, lysine, leucine, methionine, valine, isoleucine, and tryptophan. Additionally, it contains digestible carbohydrates, reducing sugars, glucose and fructose, lipids, fatty acids, phenolic compounds, vitamins, macro-elements and microelements, but also bioactive substances such as flavonoids [5]. Bee pollen has been reported as possessing antimicrobial, antifungal, anti-inflammatory, antiviral, immunostimulant and local analgesic properties [5]. These beneficial effects of bee pollen on health are a result of the diverse array of secondary plant metabolites (niacin, tocopherol, thiamine, polyphenols, phytosterols) as well as coenzymes and enzymes contained in it [6]. Bee pollen is good source of polyphenol and flavonoid compounds, which have the ability to scavenge free radicals [6]. Bee pollen contains flavonoids such as rutin, quercitrin, isoquercitrin, naringenin, kaempferol, and luteolin [7]. Flavonoids that have antiviral, antibacterial, anti-oxidation, anti-aging, anti-inflammatory, anti-tumor and analgesic activities are among the main bioactive substances in bee pollen and important indicators of the quality of bee pollen [7]. The ratio of these substances in bee pollen largely depends on factors such as soil type, meteorological circumstances, and bee race, which vary according to plant origin and geographical origin [8,9,10].

Natural antioxidants can be isolated as pure individual compounds from different products and used for food preservation, as functional/supplements food, in cosmetics, and for therapeutic purposes [11]. Numerous studies on the antioxidant capacity of foods/food supplements including bee pollen have been conducted in recent years. Many studies have demonstrated that bee products, including pollen, contain significant amounts of natural antioxidants and exhibit a high level of antioxidant activity [10,12]. Bee pollen’s antioxidant capacity varies according to the plant from which it is derived, as well as the geographical and climatic characteristics of the region in which it is provided [8,9,10]. Considering the differences that may arise from these changes, biological properties such as antioxidant activity, in particular, should be included in the standard studies to be conducted on pollen [12]. In addition, in order to determine the pollen’s quality and conduct standardization studies, it is necessary to determine the pollen’s plant origin via palynological analysis and then its chemical composition. This study was carried out by using bee pollen samples produced in Bayburt Region, which is located in the Eastern Black Sea Region of Turkey. Bayburt, an important province for beekeeping in Turkey, is located quite close to the production area of Turkey’s most expensive honey, Anzer honey. Bee products produced in the Anzer Region are sold at higher prices than bee products produced in other regions of Turkey. Despite the close proximity of Bayburt to the Anzer Region, the fact that there is such a price difference causes the producers to suffer. For this reason, this study is very important in terms of presenting the quality of the pollen produced in the region as well as that of the plant resources that provide these samples with unique characteristics. It has been clearly demonstrated in many publications that the physical, physico-chemical, chemical and bioactive properties of plant-derived bee products such as honey and pollen vary depending on variables such as geographical location, climatic characteristics, collection time and most importantly plant sources. For this reason, determining the specific properties of products produced in different regions can contribute both to standardization studies in the future and to scientists’ doing research in this field. In addition, determining the characteristics of local products can provide support to beekeepers who produce in that region to promote their products. As a matter of fact, in recent years, trust and demand for geographically indicated natural products has increased in Turkey, as in the whole world. Although numerous researchers have examined the palynological analysis of pollen samples produced in Turkey, to the best of our knowledge no research on the plant origin of pollen samples from Bayburt has been conducted. In this regard, this research has the characteristic of being a preliminary study. To contribute to standardization studies, the botanical origin, antioxidant capacity, total phenolic content, total flavonoid content, phenolic profile and fatty acid content of bee pollen extract were evaluated in this study.

## 2. Results and Discussions

### 2.1. Botanical Origin of Bee Pollen Loads

The pollen loads examined here were collected directly from hives and consisted of a mixture of pollen types from a variety of plant species, namely Asteraceae, Fabacea, Campanulaceae, Cistaceae and Rosacaeae. Numerous studies similar to ours have determined the botanical origins of bee pollen samples. Mayda et al. [10] identified different plant taxa in pollen samples collected from beehive samples in various regions of Turkey. These researchers identified Asteraceae (*Centaurea* sp., *Cichorium* sp., *Helianthus* sp., *Taraxacum* sp., *Xanthium* sp., *Senecio* sp., *Conyza* sp., *Solidago* sp., *Crepis* sp., *Bellis* sp., *Carduus* sp.), Apiaceae (*Daucus* sp.,), Berberidaceae (*Berberis* sp.), Betulacae, Boraginacea (*Echium* sp., *Myosotis* sp.), Brassicaceae, Caryophyllaceae (*Dianthus* sp.), Chenopodiaceae, Cistaceae (*Cistus* sp.), Dipsacaceae, Fabaceae (*Astragalus* sp., *Coronill*a sp., *Onobrychis* sp., *Trifolium* sp., *Lotus* sp., *Trifolium pratense*, *Melilotus* sp., *Hedysarum* sp.), Fagaceae (*Castanea sativa*), Geraniaceae, Hypericaceae (*Hypericum* sp.), Lamiaceae (*Salvia* sp., *Teucrium* sp., *Thymus* sp.), Lauracaeae (*Laurus* sp.), Myrtaceae (*Eucalyptus* sp.), Papveraceae, Plantaginaceae, Poaceae (*Zea mays*), Polyganaceae (*Rumex* sp.), Ranunculaceae, Rosaceae (*Fragaria* sp., *Linaria* sp., Pyrus sp., *Sarcopoterium* sp., *Sanguisorba* sp.), Tiliaceae (*Tilia* sp.), Loranthaceae (*Loranthus* sp.), Scrophulariaceae (*Linaria* sp.), Salicaceae (*Salix* sp.), Ericaceae (*Rhododendron* sp.) and Solanaceae taxa as the botanical origin of pollen samples from different regions of Turkey. Almeida-Muradian et al. [13] revealed that pollens from the Arecaceae, Asteraceae and Myrtaceae families are abundant in bee pollen. Feás et al. [14] discovered the presence of pollens belonging to nine taxa in samples of bee pollen from Portugal. In their studies, it was determined that bees’ most visited plants for collecting pollen were Cistaceae and Boraginaceae family members, and the least visited plants were Myrtaceae family members. In Turkey, agriculture and vegetation are different, which results in the formation of important resources for beekeeping. Therefore, the nutritional value and chemical content of bee pollen varies depending on the plant species from which it is collected [10]. The morphological structure of pollen varies significantly between plant species [15]. Therefore, it is possible to encounter a variety of pollens of all kinds of colors, shapes and surface structures [15]. Although pollen is generally yellow, it is possible to encounter pollens of various colors such as red, purple, pink, lilac, green and black [16]. Due to the bee’s approximately 5 km flight range, it is impossible to determine with a naked eye where and from which plant the bee collects pollen [16,17,18]. Therefore, pollen analysis is critical to determine the botanical sources of bee pollen obtained from different plant sources, such as those in this study.

### 2.2. Total Phenolic/Flavonoid Content and Antioxidant Capacity

All plants produce a large number of phenolic substances as secondary metabolites in their metabolism [19,20]. For this reason, there are always different amounts of various phenolic compounds in all plant-based foods. Phenolic compounds are divided into phenolic acids and flavonoids. It is known that phenolic compounds found in natural plants are effective at protecting the human body against free radicals [21]. These compounds exhibit a variety of beneficial properties such as anticarcinogenic, antioxidant, antiapoptotic, anti-atherosclerosis, anti-inflammation, cardiovascular protection, anti-aging, and enhancement of endothelial function, as well as inhibition of angiogenesis and cell proliferation activity [22]. Bee pollen also contains relatively high amounts of phenolics, and the flavonoids (0.2–2.5%) are the main compounds [23]. In this study, the total phenolic content of pollen extract from mixed pollen types was calculated spectrophotometrically using the Folin–Ciocalteu method as an equivalent to gallic acid. In addition, the total flavonoid content was calculated using the aluminum chloride method to be equivalent to quercetin. As shown in Table 1, significant quantities of total phenolic and total flavonoid content were recorded in bee pollen extract 173.52 ± 1.87 mg GAE/g and 79.21 ± 5.89 mg QE/g, respectively. The total phenolic content obtained in this study was within the range reported for Anzer pollens from Turkey (44.07–124.20 mg GAE/g) [24] and pollens from Brazil (41.5–213.2 mg GAE/g) [25], but was higher than that reported for Turkish pollen samples (7.88 mg GAE/g and 17.46 mg) [26], Italian pollens (13.53–24.75 mg GAE/g) [27], Romanian pollens (4.4–16.4 mg GAE/g) [28], and Portuguese pollens (10.5–16.8 mg GAE/g) [29]. Similarly, the flavonoid content of pollen extract has been determined in various studies in different ratios, such as 4.5–7.1 mg CAE/g, 1.81–4.44 mg/g QE/g [10] and 1.42–9.05 mg QE/g [8], but Bayburt pollen has a higher flavonoid content than these. In addition, a different study discovered that Bayburt honey contained fewer total phenolics (219.43 mg GAE/kg–768.82 mg GAE/kg) than the pollen sample [30]. The amounts of 1358 ± 90 mg GA/100 g total phenolics were reported for heterofloral Bayburt bee pollen [26], which was lower than the total phenolics of our pollens. The total phenolic and total flavonoid content of twenty-two different pollen samples from Portugal were determined as 12.9–19.8 mg GAE/g and 4.5–7.1 mg CAE/g, respectively [14]. In comparison with the literature, the pollen extract’s total phenolic and total flavonoid content is quite high, indicating that the pollen extract from Bayburt has a high antioxidant activity.

Especially in recent years, as the incidence of numerous diseases such as cancer has increased, it has become critical to determine the activities of foods or food supplements used as antioxidants. When evaluated in this respect, many techniques (CUPRAC, CERAC, DPPH, FRAP, ORAC, ABTS, etc.) are used to determine the antioxidant capacity of foods [31]. In this study, CUPRAC and CERAC methods were used to test the antioxidant capacity of bee pollen, which is one of the products proven in the literature to have antioxidant properties [32] (Table 1). The bee pollen extract analyzed in this study was found to possess significant antioxidant activity as well as a variety of phenolic and flavonoid compounds. Some authors reported similar correlations between phenolic content and antioxidant activity using various methods [33,34,35]. Anzer pollen was found to have remarkable antioxidant activity (11.77 to 105.06 µmol Trolox/g pollen) in a similar study [24]. Altıner et al. [36] investigated the antioxidant capacity of extracts of bee pollen from specific floral regions in Turkey. Altıner et al. [36] reported that antioxidant capacities of pollen samples were 83.24–257.27 μmol TE/g by the CUPRAC method.

### 2.3. Polyphenolic Profile

The number of studies on the characterization of phenolic compounds in bee pollen has increased in recent years, as phenolic compounds have been recognized as potentially useful taxonomic markers [37]. The content of bee pollen varies significantly depending on the geographical origin, harvested time and plant source [38]. While the chemical, physicochemical and sensory properties of monofloral pollen are similar, these properties are quite different for heterofloral pollen samples. In this study, the heterofloral pollen extract’s polyphenolic profile was evaluated qualitatively and quantitatively using LC-MS/MS for 23 phenolic compounds. The concentrations of individual compounds in the pollen extract are summarized in Table 2. Sixteen of them were identified in the extract based on standard retention times and mass spectra. Gallic acid, protocatechuic acid, syringic acid, 2,5-dihydroxybenzoic acid, caffeic acid, salicylic acid, catechin, rutin, *p*-coumaric acid, trans ferulic acid, phlorizin, myricetin, luteolin, quercetin, kaempferol and isorhamnetin were all present in different amounts in the pollen sample, whereas chlorogenic acid, sinapinic acid, naringin, ethyl gallate, propyl gallate, 2-hydroxytranscinnamic acid and resveratrol were present below their limits of detection. The results indicated that the extract contained a high concentration of phenolics. The major phenolic compound in bee pollen extract was rutin (115,442.25 ± 7774.28 µg/kg), followed by kaempferol, quercetin, myricetin and *p*-coumaric acid at contents of 9870.72 ± 790.14, 7849.8 ± 528.63, 2220.70 ± 177.76, 1508.98 ± 91.89 µg/kg, respectively. However, rutin, quercetin and kaempferol have been reported at high rates in many previous studies [39,40]. Rutin, which is believed to contribute to antioxidant activity [41], can be used as a chemical marker for bee pollen. However, caffeic acid, isoquercitrin, galangin and chrysin were determined as “key markers” in monofloral and heterofloral bee pollen samples from different botanical and geographical origins [42]. On the other hand, the presence of *p*-coumaric acid, ferulic acid, rutin, myricetin, quercetin and kaempferol phenolics in bee pollen samples has also been shown in previous studies [43]. However, kaempferol and myricetin phenolics were found only in low quantities in chestnut pollen samples from the Black Sea Region, Turkey [44]. This may be due to the monofloral nature of the pollen sample. As a matter of fact, in some studies, it has been determined that heterofloral bee pollen samples have richer bioactive content compared to monofloral pollen samples [42]. As in our study, the following flavonoids and their derivatives were identified as components of Morocco bee pollen: ferulic acid (17.17 mg/kg), o-coumaric acid (27.10 mg/kg), chlorogenic acid (not detected), catechin (not detected), naringin (113.71 mg/kg), quercetin (48.12 mg/kg), rutin (95.36 mg/kg), resveratrol (44.00 mg/kg), and kaempferol (not detected) [45]. In addition, Thakur and Nanda [40] reported that the concentration of flavonoids (catechin: 0.94–19.10 mg/100 g; rutin: 4.81–24.83 mg/100 g; quercetin: 3.14–15.94 mg/100 g; luteolin: 1.06–5.86 mg/100 g; kaempferol: 0.12–9.35 mg/100 g; and apigenin: 0.46–3.02 mg/100 g) varied depending on the botanical origin. In a study related to bee pollen phenolics, Ulusoy and Kolaylı (2014) [24] investigated the flavonoid content in Turkey bee pollen and determined gallic acid, protocatechuic acid, caffeic acid, rutin and quercetin phenolics. The types and concentrations of phenolic compounds in pollen samples from various geographical sources may have differed depending on the botanical origin and climatic conditions, as well as the processing and storage conditions.

### 2.4. Fatty Acid Composition

Bee pollen’s diversity of compounds makes it the most important nutrient for bees while also serving as a good food supplement for humans [9,46]. Pollen grains contain between 1% and 20% of lipid substances [9]. Around 3% of total lipids are free fatty acids, about half of which are unsaturated acids like oleic, linolenic (omega-3) and linoleic (omega-6) [12,47]. The fatty acid content varied between 0.52 and 8.21%. In this study, 13 fatty acids were identified in the pollen extract. Individual percentages of each fatty acid are given in Table 3. Cis-4,7,10,13,16,19 docosahexaenoic acid (C22:6n3) was determined as the predominant fatty acid (8.21%) in the pollen sample, and cis-11-eicosenoic acid (C20:1n9) was detected as the second most abundant fatty acid in the extract, accounting for 7.27%. In addition, alpha-linolenic acid, a source of omega-3 fatty acids, was the fourth type of fatty acid detected in the pollen extract, with a share of 5.25%. Since omega-3 and omega-6 fatty acids cannot be synthesized in the human body, they must be obtained from food [48]. In addition, alpha-linolenic acid is a precursor for the synthesis of EPA and DHA, which are not synthesized in the human body and which have many important functions in the body [48]. These findings suggest that pollen extracts can be potentially used as a source of omega-3.

The results obtained for the fatty acid profile of the pollen extract examined in this study are consistent with those obtained in previous studies in the literature, including in the works of Kostic et al. [9] and Araújo et al. [8]. In fact, Kostic et al. [9] investigated the fatty acid profile of Serbian bee-collected pollen and discovered twenty different fatty acids, with palmitic, oleic and linolenic acids being the most abundant, respectively. Likewise, Sagona et al. [49] reported on a similar fatty acid profile for monofloral organic Tuscanian bee pollen samples of different botanical origins, which contained α-linolenic acid and linoleic acid as dominant fatty acids. Mărgăoan et al. [50] detected 14 different polyunsaturated fatty acids in multifloral bee pollen samples from Romania. They detected a high concentration of α-linoleic acid (20.28–49.37%), followed by linoleic acid (7.62–33.93%), oleic acid (3.68–15.34%). In pollen samples from the Bingöl Region, Karagözoğlu et al. [51] reported that palmitic acid ranged between 24.25% and 32.49%, stearic acid ranged between 1.07% and 1.95%, oleic acid between 5.3% and 9.03%, linoleic acid between 8.56% and 10.76% and α-linolenic acid between 27.26% and 36.53%. In addition, these researchers stated that by analyzing the fat content of pollen, they could determine its biological value and nutritional quality. Compared to our study, Mărgăoan et al. [50] and Karagözoğlu et al. [51] reported higher concentrations of fatty acid content in pollen samples. These differences may be due to the fact that the pollen samples were obtained from different botanical and geographical sources, as well as the methodological differences (fatty acid extraction method/extraction solvent, etc.) used in the studies.

### 2.5. In Vitro Antibacterial Activity

The agar well diffusion (AWD) method was used in this study to determine the in vitro antibacterial activity of bee-collected pollen extracts (BCPE) against different food-borne pathogens. For this purpose, the dry residue from the prepared BCPE was weighed and dissolved in 80% DMSO. Thus, a solution containing 20 mg/mL of the extract was prepared and used to determine its in vitro antibacterial activity (Table 4). As a result, we observed that the inhibition zone diameters for Gram-positive bacteria ranged between 15–18 mm at the end of the 24 h incubation period (*Bacillus cereus* ATCC 14579: 15 mm; *Staphylococcus aureus* NCTC 10788: 18 mm). In addition, for Gram-negative bacteria, it was observed that the inhibition zone diameters varied between 9 and 12 mm *(Escherichia coli* ATCC BAA 25-23: 9 mm; *Salmonella* Typhimurium RSSK 95091: 12 mm).

Following these procedures, the broth micro-dilution technique was used to determine the BCPE sample’s minimum inhibition concentration (MIC) values against a panel of food-borne pathogens. At the end of this procedure, it was observed that the MIC values against Gram-positive microorganisms ranged from 2.5 to 5 mg/mL (for *Staphylococcus aureus* NCTC 10788, MIC value 5 mg/mL; 2.5 mg/mL for other Gram-positive bacteria). MIC values for Gram-negative bacteria were found to be between 5 and 10 mg/mL (MIC value of *Salmonella* Typhimurium RSSK 95091 5 mg/mL; for other Gram-negative bacteria, 10 mg/mL).

Graikou et al. [52] reported that Greek pollen samples have antioxidant, antimicrobial and proteasome activation properties. In addition, they stated that Gram-positive bacteria were more sensitive than Gram-negative bacteria to pollen extract. However, it should be noted that our study obtained higher MIC values than this study. Mohdaly et al. [53] investigated the antioxidant properties and antibacterial activity of pollen and propolis methanol extracts. Similarly, pollen extracts were found to be effective against both Gram-positive and Gram-negative bacteria in this study. In addition, MIC values were found to be lower in Gram-positive bacteria than in Gram-negative bacteria. When all of these findings are considered together, it becomes clear that the findings of this study are also consistent with the results obtained in our study. Additionally, Cabrera and Montenegro [54] determined the antimicrobial activity of pollen aqueous extract against *S. aureus*, *Streptococcus pyogenes*, *E. coli* and *P. aeruginosa* bacteria. These researchers found higher MIC values for pollen extracts compared to our study. One possible explanation for this observed difference is that the pollen samples were extracted with water. In conclusion, when antibacterial activity is considered, the results of this study indicate that BCPE has an inhibitory effect against food-borne pathogens at various concentrations. As can be seen, these findings are quite consistent with and corroborate those previously published in the literature.

## 3. Materials and Methods

### 3.1. Bee Pollen Loads and Determination of Their Botanical Origin

Bee pollen sample loads were collected in 2018 from Bayburt, Turkey, with bottom-fitted pollen traps. The pollen samples were stored at −18 °C until analysed. The botanical origin of the bee pollen loads was prepared according to Wodehouse [55]. A few drops of 96% ethyl alcohol were dropped on the pollen on the slide. The preparation was allowed to stand on the heater until the alcohol evaporated. A portion of basic fuchsine added glycerin-gelatin was placed on the pollen and mixed. Pollen grains were examined by a Leica DM500 light microscope and diagnosed using an immersion objective lens (×100) [17,56].

### 3.2. Preparation of Bee Pollen Extract

Bee pollen samples collected from the hive were pulverized using a grinder. After that, a 1.5 g sample was dissolved in 10 mL ethanol (95%) followed by ultrasonic-assisted extraction in an ultrasonic cleaning bath for 60 min at 40 °C. The mixture was centrifuged for 30 min at 3500 rpm and then the supernatant was collected. Extraction was repeated two more times and supernatants were combined in a flask. The extract was filtered through a 0.45 µm membrane and diluted to the appropriate concentration as per calibration curves [57].

### 3.3. Yield of Extraction

Extraction efficiency is a measure of the efficiency of the solvent and method for the extraction of phytochemical components. In the scope of our study, the extraction efficiency of bee pollen samples was calculated by making some modifications to the study conducted by Zhang et al. [58]. The solvent of the extract was removed from the Miulab NKD-200/2 model under nitrogen (40 °C, flow 1 mL/min, duration 8 min). The extract was incubated at −80 °C for 1 h and freeze dried for 24 h at −52 °C and 0.05 mbar pressure with a Christ Alpha 1-2LD model lyophilizator and the sample was weighed.


Extraction yield (%) = (weight of freeze − dried
extract × 100)/(the weight before the process)


### 3.4. Total Phenolic Content (TPC)

TPC was determined using a modified Folin–Ciocalteu method by a microplate reader (Epoch Microplate Spectrophotometer, BioTek Instruments, Inc.) using spectrophotometric detection and microtiter 96-well plates. To start, 50 µL of the extract and 50 µL of Folin-Ciocalteu reagent (1:5, *v*/*v*) were placed in each well. After that, 100 µL of sodium hydroxide solution (0.35 M) was added. The absorbance at 760 nm of the blue complex formed was monitored after 3 min. The results were expressed as gallic acid equivalent (mg GAE/g) [59].

### 3.5. Total Flavonoid Content (TFC)

Total flavonoid analysis of the extract was performed by a modified version of the method used by Jia et al. [60]. Accordingly, an aliquot (1 mL) extract was mixed with 0.3 mL 10% AlCl_3_.6H_2_O solution after the addition of 0.3 mL 5% NaNO_2_ solution. Then, 2 mL of 1 M NaOH solution was added, 2.4 mL of water was added, and the mixture was stirred. At 510 nm the absorbance was measured against the prepared reagent blank a microplate spectrophotometer (Epoch Microplate Spectrophotometer, BioTek Instruments, Inc., Winooski, VT, USA). TFC was expressed as mg quercetin equivalent (mg QE/g).

### 3.6. Antioxidant Activity

#### 3.6.1. Cupric ion Reducing Antioxidant Capacity (CUPRAC) Assay 

In a test tube, 1 mL of 1.0 × 10^−2^ M copper (II) chloride solution, 1 mL of 7.5 × 10^−3^ M neocuproine solution and 1 mL of 1 M ammonium acetate buffer (pH = 7.0) were added and mixed. Then, 0.1 mL of antioxidant sample solutions and 1 mL of distilled water were added to these solutions. The final volume of the mixture was 4.1 mL. Tubes were kept at room temperature with cover closed for 30 min. At the end of the period, the absorbances of the solutions at 450 nm were measured against the reagent blank and compared to a Trolox standard curve 1–20 mM [61].

#### 3.6.2. Ce(IV)-Based Reducing Capacity (CERAC) Assay

The total antioxidant capacity of the samples was also detected using of the Cerium (IV) assay of Ozyurt et al. [62]. This involved adding 1 mL of 2.0 × 10^−3^ M Ce(IV) solution to 1 mL of extract. Then, the mixture was diluted to 10 mL with distilled water and the solution was allowed to stand for 30 min at room temperature. The absorbance of the mixture was measured at 320 nm against a reagent blank composed of distilled water and compared to a Trolox standard curve 1–20 mM.

### 3.7. Liquid Chromatography-Triple Quadrupole Mass Spectroscopy (LC-MS/MS) Analysis 

Extraction was carried out via modified methods of isolation of phenolic compounds developed by Fischer et al. [63]. First, 100 µL of extract was mixed with 900 μL extraction solution and, afterward, the samples were vortexed for 30 s. After that, the mixture was homogenized using a sonicator (WiseClean, DAIHAN) for 10 min, centrifuged for 5 min at 3500 rpm, and the clear supernatant was injected into the LC–MS/MS system for analysis. 

### 3.8. Fatty Acid Methyl Ester Analysis

#### 3.8.1. Lipid Extraction

The bee pollen sample was freeze-dried for approximately 24 h to a constant mass and ground to a homogenous powder using a coffee blender. Soxhlet extraction was performed in a BUCHI Extraction System B-811 (Buchi, Switzerland). Soxhlet extraction of the bee pollen (2 g) was performed with 100 mL hexane for approximately 8 h. 

#### 3.8.2. Methylation Procedure

Fatty acid methyl esters were prepared with some modifications following the IUPAC methodology. A 100 mg sample of pollen oil was dissolved in 10 mL n-hexane (GC grade). A methanolic potassium hydroxide (KOH) solution (2 N) was added (0.1 mL). The tube was sealed and mixed vigorously for 120 s in a vortex shaker. Saturated sodium chloride (NaCI) solution (2 mL) was added, and the organic phase was separated. An aliquot (1 µL) of the hexane solution was injected to GC analysis [64]. 

#### 3.8.3. Gas Chromatography-Mass Spectroscopy (GC-MS) Conditions

Methylated fatty acid samples were analyzed by Agilent 6890A GC gas chromatography and 5973C MSD mass spectrometry. The components were separated in a fused silica capillary column DB-23 (60 m × 0.25 mm ID, 0.15 µm; J & W 122-2361). The oven temperature was maintained at 50 °C for 1 min, 25 °C increments at 175 °C, and 4 °C increments at 230 °C for 5 min. The injection temperature was set to 230 °C. A 1 µL injection was made and the split ratio was adjusted to 1:50. Helium was used as a carrier gas with a flow rate of 1 mL/min. The injector and detector temperatures were 240 °C and 260 °C, respectively. The mass spectrometer was operated in the electron impact (EI) mode at 70 eV in the scan range of 50–550 *m*/*z*. Peak identification of the fatty acids in the analyzed bee pollen sample was carried out by the comparison with retention times and mass spectra of FAME component mix [64].

### 3.9. Determination of Antibacterial Effect

#### 3.9.1. In Vitro Antibacterial Activity of BCPE

In order to determine in vitro antibacterial activity of bee-collected pollen extract (BCPE), the agar well diffusion method (AWD) was applied. In this process, 5 Gram-positive bacteria and 7 Gram-negative bacteria were used (Table 4). For this purpose, initially, 40 mg of the dried residue BCPE was weighed and transferred to a 2 mL Eppendorf tube (BCPE was prepared as described above). Afterwards, 2 mL of dimethyl sulfoxide (DMSO) was added to this tube and vortexed. After that, the extract was thoroughly dissolved in this solvent. Microorganisms kept as stock cultures at −20 °C were transferred to a Mueller–Hinton agar (MHA) medium via a sterile loop and incubated at 37 °C for 24 h. At the end of the 24 h incubation period, a loop full sample was taken from the single colonies and transferred to Mueller–Hinton broth (MHB) medium in 15 mL falcon tubes and incubated again at 37 °C for 18 h. After incubation of microorganisms in liquid media (MHB), turbidity was adjusted to 0.5 McFarland standard turbidity and these bacteria suspensions were used for determination of antibacterial activity. Immediately after these procedures, 8 mm diameter wells were prepared on MHA media using a sterile cork-borer. Samples of pathogenic microorganisms in broths set to 0.5 McFarland standard turbidity were spread onto MHA media to cover the entire surface with a sterile swab. After this process, 50 µL of the BCPE (at a concentration of 20 mg/mL) was taken and transferred to the wells using a micropipette. After these procedures were completed, the petri dishes were incubated at 37 °C for 24 h. At the end of the incubation period, the inhibition zones observed around the wells were recorded by measuring with the help of a ruler. In addition, 80% DMSO was used for negative control [65,66].

#### 3.9.2. Broth Micro-Dilution Method

The broth micro-dilution method was used to determine the minimum inhibition concentration (MIC) values of the BCPE. Initially, 95 µL of sterile nutrient broth were transferred to each well of the 96-well polystyrene microplate using a multichannel pipette. Immediately after this procedure, 5 µL of the pathogenic microorganism sample was transferred to all wells (at 0.5 McFarland standard turbidity). After this inoculation process, 100 µL of the BCPE was transferred to only the first wells (at a concentration 20 mg/mL). Thus, 200 µL suspensions were prepared only in the first wells and 100 µL in the other wells. After these procedures were completed, the mixture in the first wells was thoroughly mixed with a micropipette and then 100 µL of suspension was taken from the first wells and transferred to the second wells. After this suspension was mixed thoroughly, 100 µL samples from the second wells were transferred to the third wells and these procedures were repeated until the last well (8th well). Thus, starting from the first well, the BCPE concentration was diluted by half in all wells consecutively [52,67].

## 4. Conclusions

Regular consumption of bee pollen is recommended in many countries, including Turkey. Therefore, to reveal the quality and physicochemical properties of the bee pollen products is of great importance for standardization studies. As a result of this study, we have determined that the mixed pollen loads from Bayburt have a high total phenolic and total flavonoid content and high antioxidant activity in parallel. The results obtained in this study revealed that the Bayburt pollen extract contained high amounts of individual phenolic compounds, such as rutin, kaempferol and quercetin. Therefore, it can be said that Bayburt bee pollen is a natural source of antioxidants. Moreover, the results obtained showed that a bee pollen sample from this region can be used as an important source of fatty acids, such as cis-4,7,10,13,16,19 docosahexaenoic acid, cis-11-eicosenoic acid, palmitic acid, and alfa linolenic acid. As a result, the fact that the bioactive content of Bayburt pollen extract is quite remarkable shows that it can be used as a dietary supplement.

## Figures and Tables

**Table 1 molecules-27-00117-t001:** Antioxidant properties of pollen extract.

TPC (mg GA/g)	TFC (mg QE/g)	CUPRAC (mg Trolox/g)	CERAC (mg Trolox/g)
173.52 ± 1.87	79.21 ± 5.89	85.59 ± 5.95	118.13 ± 19.90

**Table 2 molecules-27-00117-t002:** Concentrations of polyphenolic compounds in pollen extract (µg/kg).

Compounds	Retention Time (min)	RSD (%)	[M−H]^−^ *m*/*z*	Ion Pair	R^2^ (Linearity)	Concentrations (µg/kg)
Gallic acid	1.552	0.482	168.9	168.9/125; 168.9/78.8	0.9983	585.52 ± 31.84
Protocatechuic acid	1.830	0.482	153.1	153.1/109.1; 153.1/90.8	0.9980	441.94 ± 21.13
2,5-Dihydroxybenzoic acid	2.084	0.482	152.9	152.9/107.9; 152.9/53.1	0.9966	58.80 ± 3.95
Caffeic acid	3.579	0.488	179	179/135.1; 179/117.3	0.9983	928.56 ± 74.33
Syringic acid	3.603	0.489	196.9	196.9/182.1; 196.9/121.1	0.9967	284.24 ± 17.30
Salicyclic acid	3.691	0.577	136.8	136.8/93.1; 136.8/65	0.9947	314.11 ± 4.39
Chlorogenic acid	3.701	0.488	352.9	352.9/191; 352.9/82	0.9982	Nd
Catechin	3.888	1.155	288.9	288.9/245; 288.9/205	0.9971	37.26 ± 1.78
Rutin	3.911	0.494	609	609/299.9; 609/270.9	0.9975	115442.25 ± 7774.28
Sinapic acid	3.924	0.417	222.9	222.9/208; 222.9/120.9	0.9946	Nd
*p*-Coumaric acid	3.986	0.551	163.1	163.1/118.9; 163.1/93	0.9972	1508.98 ± 91.89
Naringin	4.030	0.491	579.1	579.1/458.9; 579.1/271	0.9990	Nd
Trans ferulic acid	4.045	0.442	193	193/177.9; 193/134.1	0.9982	151.84 ± 8.25
Ethyl gallate	4.078	0.492	197	197/169; 197/124	0.9971	13.45 ± 0.64
Phlorizin	4.081	0.540	434.1	434.8/272.9; 434.8/167	0.9977	Nd
Myricetin	4.167	0.058	317	317/178.8; 317/150.9	0.9961	2220.70 ± 177.76
Propyl gallate	4.171	0,539	211	211/124.1; 211/78	0.9963	Nd
2-Hydroxytranscinnamic acid	4.173	0,477	162.9	162.9/119; 162.9/92.8	0.9972	Nd
Resveratrol	4.314	0.491	226.9	226.9/184.9; 226.9/142.8	0.9983	Nd
Luteolin	4.321	0.981	284.9	284.9/150.9; 284.9/133	0.9966	250.47 ± 11.97
Quercetin	4.346	0.473	301	301/178.9; 301/150.9	0.9964	7849.8 ± 528.63
Kaempferol	4.406	0.956	284.9	284.9/226.9; 284.9/93	0.9985	9870.72 ± 790.14
Isorhamnetin	4.421	0.489	314.9	314.9/299.9; 314.9/151	0.9980	523.545 ± 31.88

Nd: not detected.

**Table 3 molecules-27-00117-t003:** Fatty acid composition of pollen sample.

Fatty Acids	Formula	Retention Time	% of Total
Caprylic acid	C8:0	5.022	0.52 ± 0.03
Capric acid	C10:0	6.118	2.89 ± 0.08
Lauric acid	C12:0	7.191	1.53 ± 0.01
Myristic acid	C14:0	8.531	0.74 ± 0.03
Palmitic acid	C16:0	10.382	5.5 ± 0.31
Stearic acid	C18:0	12.789	1.72 ± 0.05
Oleic acid	C18:1n9t	13.121	3.95 ± 0.03
Linoleic acid	C18:2n6c	13.793	3.1 ± 0.13
Alfa linolenic acid	C18:3n3	14.662	5.25 ± 0.30
Cis-11-Eicosenoic acid	C20:1n9	15.959	7.27 ± 0.21
Erucic acid	C22:1n9	18.936	5.54 ± 0.04
Nervonic acid	C24:1n9	21.728	2.21 ± 0.09
Cis-4,7,10,13,16,19 Docosahexanoic acid	C22:6n3	21.892	8.21 ± 0.46

**Table 4 molecules-27-00117-t004:** In vitro antibacterial activity of BCPE (at a concentration of 20 mg/mL) via agar well diffusion (AWD) method. (Inhibition zone diameter (IZD: mm) and minimum inhibition concentration (MIC—mg/mL). For negative control (NC), 80% DMSO was used.)

	Microorganisims	BCPE		NC
		IZD	MIC	IZD
** *Gram positive* **	*Bacillus cereus* ATCC 14579	15	2.5	-
	*Bacillus cereus* BC 6830	16	2.5	-
	*Staphylococus aureus* ATCC 25923	18	2.5	-
	*Staphylococus aureus* BC 7231	17	2.5	-
	*Staphylococus aureus* NCTC 10788	18	5	-
** *Gram negative* **	*Escherichia coli* ATCC BAA 25-23	9	10	-
	*Escherichia coli* BC 1402	10	10	-
	*Escherichia coli* NCTC 9001	10	10	-
	*Pseudomonas aeruginosa* ATCC 9070	10	10	-
	*Pseudomonas aeruginosa* NCTC 12924	10	10	-
	*Salmonella typhimurium* RSSK 95091	12	5	-
	*Yersinia enterocolitica* ATCC 27729	11	10	-

## Data Availability

The data presented in this study are available on request from the corresponding author.

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
