# Peer review of "Screening of Plant Pollen Sources, Polyphenolic Compounds, Fatty Acids and Antioxidant/Antimicrobial Activity from Bee Pollen"

_molecules, 2021, doi:10.3390/molecules27010117_

Round 1

Reviewer 1 Report

I think the topic of this article is very interesting.

I suggest the following minor revision:

Page 1 line 21 I suggest writing the numbers as words when starting the sentence. Check throughout the text.

Page 1 lines 35-37 lack of bibliographic reference. Please add reference.

Page 1 line 41 please add royal jelly in the list of bee products.

Page 2 lines 60-62, 64-65, 68-69 lack of bibliographic reference. Please add reference.

Page 2 line 97 I suggest to add “at the best of our knowledge”.

Page 3 line 127 I suggest to change “diverse” with “different”.

Page 3-4 lines 128-129, 132-134, 137, 137-138, 139-140, 140, 150-152, 178-179, 179-180, 184, 185-186 lack of bibliographic reference. Please add reference.

Page 4 line 189 positive or negative correlation?

Page 9 line 349 please add the model and brand of the spectrophotometer.

Page 10 line 373 please add at which frequency the sonicator was set.

Page 10 line 405 Please write “in vitro” in italics.

Author Response

Dear Reviewers,

The queries on our paper were responded as below.

Comment 1: I suggest writing the numbers as words when starting the sentence. Check throughout the text.

Response 1: It has been checked and corrected throughout the manuscript.

Comment 2: Page 1 lines 35-37 lack of bibliographic reference. Please add reference.

Response 2: Bibliographic references have been added to the fields you specified.

Comment 3: Page 1 line 41 please add royal jelly in the list of bee products.

Response 3: Royal jelly had been added to the list.

Comment 4: Page 2 lines 60-62, 64-65, 68-69 lack of bibliographic reference. Please add reference.

Response 4: Bibliographic references have been added to the field.

Comment 5: Page 2 line 97 I suggest to add “at the best of our knowledge”.

Response 5: It was corercted.

Comment 6 : Page 3 line 127 I suggest to change “diverse” with “different”.

Response 6: It was corrected

Comment 7: Page 3-4 lines 128-129, 132-134, 137, 137-138, 139-140, 140, 150-152, 178-179, 179-180, 184, 185-186 lack of bibliographic reference. Please add reference.

Response 7: Bibliographic references were added in manuscript and all corrections were marked in red in the manuscript.

Comment 8: Page 9 line 349 please add the model and brand of the spectrophotometer.

Response 8:  They were added.

Comment 9:. Page 10 line 405 Please write “in vitro” in italics.

Response 9: It was corrected

Reviewer 2 Report

Dear authors,

Your Manuscript has potential to be interesting for researchers which analyze pollen, some future readers or potential pollen consumers.

Assessment of botanical origin of pollen sample, as well as its detailed phenolic and fatty acid quantification using advanced chromatographic techniques, that is, evaluation of some in vitro antioxidant and anti-bacterial activity can be useful for further application of pollen or its bioactive compounds in food or pharmaceutical industry.

However, there are some corrections/improvements that must be done before any further Manuscript processing.

I will enumerate different requirements to improve the manuscript. All my comments are listed below and separated as General and specific comments.

General comments:

  1. The manuscript is well organized, but several sentences are confuse and too long which makes understanding of text difficult, especially in the "Introduction" and "Results and discussion" sections. In addition, some parts of the "Introduction" are consisting of several sentences that can be shortened and concisely merged into one meaningful sentence. In that sense, suggest to authors to check a whole Manuscript once again and to rephrase/rewrite the mentioned sentences.
  2. In the whole manuscript there are too many repetitions of the same sentences and some of the quoted claims are questionable, which should be further considered and rewrite.
  3. "Abstract" should be supplemented with the main remarks obtained for anti-bacterial activity of pollen sample.
  4. The whole "Introduction" section is unordered and has no traceability of sentences. Current version is inappropriate and should be completely rewritten.
  5. The aim of this study is too local and there is no inters for broad readers all around the world. Authors should be emphasise scientific importance of research not geographic origin of sample. Moreover, the major remarks of this study are not highlighted in an appropriate form.
  6. Conclusion lacks in an adequate description of the results obtained and finally the significance of this study. Current version is too general and should be more specified. In addition, it is weird to cite previously published reference in Conclusion. Please remove reference from Conclusion because it is your results not from some other.

  7. Some references cited in the manuscript are too old, and should preferably be replaced with newer references.
  8. In Tables 2 and 3, numerical values should not be given with "," but with point "." (for examples, "585,52" replace with "585.52"). Please correct this issue.
  9. Authors should perform additional literature search and include more novel reference(s). In particular, phenolic profile should be compared with literature data. Find additional published research articles about bee pollen phenolic profiles. It is one of the most exploited theme in last decade.

Specific comments:

Line 23-25: This sentence ("The findings may ... pollen extracts") should be rewritten. What is the meaning of the term " to standardize pollen extracts" in this sentence? I think that this term is not appropriate and needs to be removed from this sentence and from the whole manuscript.

Line 32-35: Both sentences ("The fact that … more popular (Coskun, 2005)") are confused and it must be clarified. It is hard to understand what authors are trying to say here. Please, check and rewrite again.

Line 37: The term "products produced by honey bees" is unappropriated? It must be replaced with some more appropriate synonym.

Line 37-41: Sentences from this part (“The product produced … bee venom and apilarnil (Schmidt, 1996)”) can be shortened and concisely merged into one meaningful sentence.

Line 41-44: Sentences from this part (“Field bees collect … as pollen loads (Komosinska-Vassev et al., 2015).”) can be shortened and concisely merged into one meaningful sentence.

Line 44-48: Sentences from this part (“If the flora … into a receptacle (Almeida Muradian, et al., 2005)”) can be shortened and concisely merged into one meaningful sentence.

Line 48-51: Both sentences ("After bee pollen … more popular (Coskun, 2005)") are confused and it must be clarified. It is hard to understand what authors are trying to say here. Please, check and rewrite again.

Line 64-68: These sentences are not relevant for this manuscript. Please check and rewrite again or remove these sentences from the manuscript.

Line 69-71: I suggest to authors to replace part of sentence "and their significant radical cleaning capacity" with "which have ability to scavenging free radicals”. I think it is much more appropriate in this context.

Line 73-74: Remove this claim “significant amounts of nucleic acid (especially RNA)” from sentence.

Line 78: A whole sentence (“The minimum and maximum values for these substances differ significantly”) is confused and must be rewrite. Please, check and correct. Line 87-88: Rephrase and rewrite sentence (“Additionally, information about … directly related to composition”)

Line 88-92: It is desirable to remove both sentences (“Due to high … standardized products”), because the focus of this manuscript is not on toxic elements or allergens, they have not even been analyzed, so there is no need to describe them in the introduction section.

Line 92: What is the meaning of the term "Standardized products" in this sentence?

Line 124-126 and 132-134: Rephrase and rewrite sentences (“Their study … preferred taxa”) and (“Pollen varies …which it is collected”)

Line 129-131: This sentences (“Accordingly … flower bee relationship”) can be merged into one meaningful sentence.

Line 137-138: Facts in this sentences are repeated several times throughout the manuscript. Please uniform it in the whole manuscript.

Line 142-145: This whole sentence ("For this reason, … in this study.") is confused and it must be clarified. It is hard to understand what authors are trying to say here. Please, check and rewrite again.

Line 147-148: Rephrase and rewrite sentence (“Phenolic compounds … and their products”). Also, in the same sentence, term “nutritional “ replace with term “functional”. Phenolics are not nutrients but functional compounds. Please be careful with this observation.

Line 155-157: Rephrase and rewrite sentence (“Bee pollen is … are flavonoid glycosides”). In the same sentence correct following phrase “Bee pollen contains phenolic acid derivatives…" not " Bee pollen is primarily composed of phenolic acid". Phenolics in pollen are minor components. Major components are proteins, carbohydrates and lipids. Please be careful with this.

Line 171-173 and 175-177: Rephrase and rewrite sentences. You did not determine correlation between phenolic parameters and antioxidant activity. This is a strong statement, which requires a correlation analysis.

Line 178-183: These sentences are not relevant for this part of manuscript. Please delete it or rewrite and move in the Introduction section.

Line  184-187: Sentences are confused and must be rewrite. Please, check and correct.

Line 189: In this manuscript you did not perform correlation analysis. Please, conduct additional statistical analysis.

Line 196-199: These sentences should not be in this subchapter, but should be inserted in the section above (lines 146-177).

Line 213: I suggest to authors to replace term " but" with "following". Also, replace term "were also abundent" with "in the content of". I think it is much more appropriate in this context.

Line 216-220: Sentences are confused and must be rewrite. Please, check and correct.

Line 234-235: A whole sentence (The physicochemical … fatty acid esters”) is confused and must be rewrite. Please, check and correct.

Line 235: I suggest to authors to replace term "fat" with "fatty acids". I think it is much more appropriate in this context.

Line 238: I suggest to authors to replace term "oil" with "pollen sample, following cis-11-eicosenoic acid ….". I think it is much more appropriate in this context.

Line 241: I suggest to authors to replace term "at 5.25%" with "with share of 5.25%".

Line 242: What is the meaning of the term "be taken externally" in this sentence? In the same sentence, I suggest to authors add “pollen extract can be POTENTIALLY used as a source…”.

Line 254-256: A whole sentence (“Mărgăoan et al. (2014) described … of 14 polyunsaturated fatty acids.”) is confused and must be rewrite. Please, check and correct.

Line 258-263: Are the results of this study consistent with previously published results or not? Please, check and rewrite.

Line 270-271: This part of sentence “When we evaluated using the AWD method” is inappropriate. Please, check and correct.

Line 285-286: A whole sentence (“Graikou et al. (2011) analysed Greek pollen samples … activation properties.”) is confused and must be rewrite. Please, check and correct.

Line 288: A whole sentence (“Our findings appear to be consistent with those of that study”) is inappropriate and must be rewrite. Please, check and correct.

Line 292-294; 296-299; and 300-303: Sentences are confused and must be rewrite. Please, check and correct.

Line 321-323: Sentence (“Mixed bee pollen … cleaning bath for 60 min at 40 °C.”) is confused and must be rewrite. Please, check and correct.

Line 325-327:  Sentence is confused and must be rewrite. Please, check and correct.

Line 345-351: Sentences in this method are confused and must be rewrite. Please, check and correct.

Line 372: This part of sentence "Then solution was then" is unappropriated? It must be replaced. Please, check and correct.

Line 379-381: Sentences in this method are confused and must be rewrite. Please, check and correct.

Kind regards.

Author Response

REVIEWER 2:

Thank you for your constructive criticism. We understand your reserves. We tried to improve Manuscript and clarify the observed issues as much as it was possible. We hope that reviewer will understand some of our limitations that we had during this research.

Comment 1: This manuscript requires major English language copyediting.

Response1: The manuscript has been carefully reviewed by an experienced editor whose first language is English and who specializes in editing papers written by scientists whose native language is not English. A review certificate is also attached.

Dear Reviewer, first of all, this study was carried out by using bee pollen samples produced in Bayburt Region, which is located in the Eastern Black Sea Region of Turkey. Bayburt, an important province in terms of beekeeping in Turkey, is located quite close to the production area of Turkey's most expensive honey, Anzer honey.  Honey produced in the Anzer region is sold for approximately 10 times the price of honey produced in other regions of Turkey. Despite its close proximity to the region, the fact that there is such a price difference causes the producers to suffer. For this reason, this study is very important in terms of presenting the quality of the pollen produced in the region and also the plant resources that provide these samples with unique characteristics.  It has been clearly demonstrated in many literature that the physical, physico-chemical, chemical and bioactive properties of plant-derived bee products such as honey and pollen vary depending on variables such as geographical location, climatic characteristics, collection time and most importantly plant sources. For this reason, determining the specific properties of products produced in different regions can contribute both to standardization studies in the future and to scientists doing research in this field. In addition, determining the characteristics of local products can provide support to beekeepers who produce in that region to promote their products. As a matter of fact, in recent years, trust and demand for geographically indicated natural products has increased in Turkey, as in the whole world. Therefore, presenting the characteristics of local products in this way is considered as remarkable work in many respects. Although this study is a preliminary study using Bayburt pollen, it is planned to expand this study further with new researches in line with the opportunities to be obtained and to contribute to the specified areas. Although it is desired to present the planned studies together with this study, our current infrastructure facilities are suitable for completing only this much of the study for now. The publication/acceptance of our present work will provide us with new financial resources for future research.

Comment 2:  Line 23-25: This sentence ("The findings may ... pollen extracts") should be rewritten. What is the meaning of the term " to standardize pollen extracts" in this sentence? I think that this term is not appropriate and needs to be removed from this sentence and from the whole manuscript.

Response 2:  It is corrected

Comment 3:  Line 32-35: Both sentences ("The fact that … more popular (Coskun, 2005)") are confused and it must be clarified. It is hard to understand what authors are trying to say here. Please, check and rewrite again.

Response 3: Dear Reviewer, we have revised our title as follows:

“With the understanding that natural/functional foods have a positive effect in the prevention and treatment of diseases, these foods have become more popular among the public.”

Comment 4: Line 37: The term "products produced by honey bees" is unappropriated? It must be replaced with some more appropriate synonym.

Response 4: Dear Reviewer, we have revised this part as follows:

“The bee products”

Comment 5: Line 37-41: Sentences from this part (“The product produced … bee venom and apilarnil (Schmidt, 1996)”) can be shortened and concisely merged into one meaningful sentence.

Response 5: Dear Reviewer, we have revised sentences as follows:

“The products produced by honey bees are divided into two groups: the products obtained from the body secretions of the bees or directly from the bee itself (royal jelly, beeswax, bee venom and apilarnil), and the products produced by the bee using plant sources (honey, pollen, bee bread, propolis)”

Comment 6: Sentences from this part (“Field bees collect … as pollen loads (Komosinska-Vassev et al., 2015).”) can be shortened and concisely merged into one meaningful sentence.

Response 6: This part was correcetd in manuscript.

Comment 7: Sentences from this part (“If the flora … into a receptacle (Almeida Muradian, et al., 2005)”) can be shortened and concisely merged into one meaningful sentence.

Response 7: It was corrected

Comment 8: Line 48-51: Both sentences ("After bee pollen … more popular (Coskun, 2005)") are confused and it must be clarified. It is hard to understand what authors are trying to say here. Please, check and rewrite again.

Response 8: It was corrected

Comment 9: These sentences are not relevant for this manuscript. Please check and rewrite again or remove these sentences from the manuscript.

Response 9: Sentences was rewritten.

Comment 10: Line 69-71: I suggest to authors to replace part of sentence "and their significant radical cleaning capacity" with "which have ability to scavenging free radicals”. I think it is much more appropriate in this context.

Response 10:. The sentence has been modified as you suggested.

Comment 11: Line 73-74: Remove this claim “significant amounts of nucleic acid (especially RNA)” from sentence

Response 11: It is removed.

Comment 12: Line 78: A whole sentence (“The minimum and maximum values for these substances differ significantly”) is confused and must be rewrite. Please, check and correct.

Response:  It is corrected.

Comment 13: Line 124-126 and 132-134: Rephrase and rewrite sentences (“Their study … preferred taxa”) and (“Pollen varies …which it is collected”)

Response 13: It was corrected as following:

“In their studies, it was determined that the most visited plants to pollen collect by bees were Cistaceae and Boraginaceae family members, and the least visited plants were Myrtaceae family members.”

"“The nutritional value and chemical content of bee pollen varies depending on the plant species from which it is collected.”

Comment 14: Line 129-131: This sentences (“Accordingly … flower bee relationship”) can be merged into one meaningful sentence

Response 14: It was corected

Commente 15: Line 137-138: Facts in this sentences are repeated several times throughout the manuscript. Please uniform it in the whole manuscript.

Response 15: It was corrected

Comment 16: Line 142-145: This whole sentence ("For this reason, … in this study.") is confused and it must be clarified. It is hard to understand what authors are trying to say here. Please, check and rewrite again

Response 16: It was corrected as following:

“Therefore, pollen analysis is critical to determine the botanical sources of bee pollen obtained from different plant sources, such as those in this study”

Comment 17: Line 147-148: Rephrase and rewrite sentence (“Phenolic compounds … and their products”). Also, in the same sentence, term “nutritional “ replace with term “functional”. Phenolics are not nutrients but functional compounds. Please be careful with this observation.

Response 17: It was rewrite as following:

All plants produce a large number of phenolic substances as secondary metabolites in their metabolism (Fabre et al., 2001; Khoddami et al., 2012). For this reason, there are always different amounts of various phenolic compounds in all plant-based foods.

Comment 18: Rephrase and rewrite sentence (“Bee pollen is … are flavonoid glycosides”). In the same sentence correct following phrase “Bee pollen contains phenolic acid derivatives…" not " Bee pollen is primarily composed of phenolic acid". Phenolics in pollen are minor components. Major components are proteins, carbohydrates and lipids. Please be careful with this.

Reponse 18: It was corrected as following:

“Bee pollen contains also relatively high amounts of polyphenols and the flavonoids are the main compounds, most of them occur in pollen as glycosides”.

Comment 19: Line 171-173 and 175-177: Rephrase and rewrite sentences. You did not determine correlation between phenolic parameters and antioxidant activity. This is a strong statement, which requires a correlation analysis.

Response 19: It was corrected in manuscript.

Comment 20: Line 178-183: These sentences are not relevant for this part of manuscript. Please delete it or rewrite and move in the Introduction section.

Response 20: It was removed from manuscript.

Comment 21: Line  184-187: Sentences are confused and must be rewrite. Please, check and correct.

Response 21: It was rewritten.

Comment 22: These sentences should not be in this subchapter, but should be inserted in the section above (lines 146-177).

Response 22: It was corrected.

Comment 23: I suggest to authors to replace term " but" with "following". Also, replace term "were also abundent" with "in the content of". I think it is much more appropriate in this context.

Response 23: It was replaced.

Comment 24: Line 216-220: Sentences are confused and must be rewrite. Please, check and correct.

Response 24: It was written.

Comment 25: Line 234-235: A whole sentence (The physicochemical … fatty acid esters”) is confused and must be rewrite. Please, check and correct.

Response 25: It was corrected

Comment 26: Line 235: I suggest to authors to replace term "fat" with "fatty acids". I think it is much more appropriate in this context.

Response 26: It was corrected as fatty acids.,

Comment 27: Line 238: I suggest to authors to replace term "oil" with "pollen sample, following cis-11-eicosenoic acid ….". I think it is much more appropriate in this context.

Response 27: It was corrected as suggested.

Comment 28: Line 241: I suggest to authors to replace term "at 5.25%" with "with share of 5.25%".

Response 28: It was corrected “at 5.25%" with "with share of 5.25%”.

Response 29: It was corrected as suggested.

Comment 30: Line 254-256: A whole sentence (“Mărgăoan et al. (2014) described … of 14 polyunsaturated fatty acids.”) is confused and must be rewrite. Please, check and correct.

Response 30: The sentence was corrected.

Comment 31: Line 258-263: Are the results of this study consistent with previously published results or not? Please, check and rewrite

Response 31: It was rewritten.

Comment 32: Line 270-271: This part of sentence “When we evaluated using the AWD method” is inappropriate. Please, check and correct.

Response 32: It was corrected.

Comment 33: Line 285-286: A whole sentence (“Graikou et al. (2011) analysed Greek pollen samples … activation properties.”) is confused and must be rewrite. Please, check and correct.

Response 33: The sentences were rewritten.

Comment 34: Line 288: A whole sentence (“Our findings appear to be consistent with those of that study”) is inappropriate and must be rewrite. Please, check and correct.

Response 35: The sentences were rewritten.

Comment 35: Line 321-323: Sentence (“Mixed bee pollen … cleaning bath for 60 min at 40 °C.”) is confused and must be rewrite. Please, check and correct.

Line 325-327:  Sentence is confused and must be rewrite. Please, check and correct.

Line 345-351: Sentences in this method are confused and must be rewrite. Please, check and correct.

Line 372: This part of sentence "Then solution was then" is unappropriated? It must be replaced. Please, check and correct.

Line 379-381: Sentences in this method are confused and must be rewrite. Please, check and correct.

Response 35: The sentences were checked and rewritten in manuscript. All changes are marked in color in the manuscript.

Round 2

Reviewer 2 Report

Dear authors,

After firs revision your manuscript is significantly improved, however, there are additional corrections/improvements that must be done before any further Manuscript processing.

I will enumerate different requirements to improve the revised version of the manuscript. All my comments are listed below and separated as General and specific comments.

General comments

  1. The "Introduction" section is revised, however this section needs to be further rearranged and supplemented with adequate literature facts. This section should highlight the importance of the work and provide a detailed review of previous literature. I suggest authors to additionally consider the following items in the introduction:
  • Has anyone analyzed pollen or some bee products originated from Bayburt so far...
  • If pollen from Bayburt has not been analyzed, then indicate its importance (as you pointed out to me in the cover letter)...
  • Give focus to bee pollen analyzed not just in Turkey but from the world in general...
  • Give a brief overview of the bioactive compounds of pollen in the introduction with concrete facts, because the analysis of phenolic compounds of pollen has been in focus in recent years...
  1. The phenolic profile of pollen sample from Bayburt is a key item of this paper, because the health promotion, antioxidant and antimicrobial properties of the analyzed pollen potentially depend on the phenolic components. In addition, phenolic compounds are potential markers of botanical and geographical origin of pollen from the Bayburt region. Therefore, the authors must pay special attention to the subsection "2.3. Polyphenolic profiles", and to compare the retained results with recent literature data. Some recent published papers focusing on polyphenolic profiles of different monofloral or polyfloral pollen samples are listed in the appendix. I suggest to authors to consider the proposed literature and use in the interpretation of your results:

https://doi.org/10.1016/j.fct.2019.110831

https://doi.org/10.1016/j.arabjc.2021.103004

https://doi.org/10.1016/j.lwt.2019.06.011

https://doi.org/10.1016/j.lwt.2021.110973

https://doi.org/10.3390/antiox10071091

https://doi.org/10.1016/j.lwt.2019.05.105

https://doi.org/10.1021/acs.jnatprod.8b00945

https://doi.org/10.1016/j.foodres.2020.110041

https://doi.org/10.1016/j.foodres.2020.109802

  1. Conclusion lacks in an adequate description of the results obtained and finally the significance of this study. Current version is too general and should be more specified. In addition, it is weird to cite previously published reference in Conclusion. Please remove reference from Conclusion because it is your results not from some other.
  2. Style for references citing in the text is not in accordance with MDPI rules. Namely, references in the text should be listed numerically and not by the name of authors. You should correct the whole Manuscript accordingly.

Specific comments

Line 25-26: This sentence („ It was observed that ... 9-12mm for Gram negative bacteria“) must be deleted from abstract. Please, check and correct.

Line 37: „point (.)“ must be after reference („...among the public (CoÅŸkun, 2005).“)

Line 40: The term "products produced by honey bees" is unappropriated? It must be replaced with some more appropriate synonym. For examples („Honey bees products are divided ...“)

Line 60: Term “due to “ replace with term “depends of“. Please, check and correct.

Line 62-63: This sentence („These variations also... bioactive properties“) must be deleted. Please, check and correct.

Line 65: This sentence („For these reasons, ... dietary supplement (Denisow and Denisow-65 Pietrzyk, 2016)“) must be deleted, because it is unappropriate in this part of introduction.

Line 66-68: This sentence („Bee pollen bioactive  ... free radicals“) is unappropriate. For examples, this sentence can be write:"Bee pollen’s is good source polyphenol and flavonoid compounds, which have ability to scavenging free radicals".

Line 68-70: This sentence („Many studies have ... exhibit a high level of antioxidant activity (Campos et al., 2010; Mayda et al., 2020)“) must be before sentence ((„Bee pollen bioactive  ... free radicals“), line 66-68).

Line 70-73: These sentence („Antioxidants are critical for... from food (Karabulut and Gülay, 2016).“), must be deleted. This is unappropriate in this part of introduction....

Line 135-136: This sentence („Bee pollen contains also relatively high ... in pollen as glycosides.“) is confused and should be rewritten.

Line 140-1412: This sentence („As shown in ... flavonoid substances, respectively“) should be rewritten.

Line 142: Term “presented in Table 1 “ replace with term “obtained in this study“. Please, check and correct.

Line 153-155: This sentence („Feás et al. (2012) examined 22 bee ... from 12.9-19.8 mg GAE/g to 4.5-7.1 mg CAE/g“) should be rewritten. Please, check and correct.

Line 154 and 156: What is the meaning of the term "phenolic-flavonoid" in these sentences? Please, replace with adequate synonyms.

Line 164: At the end of sententce („... in the literature (Okan et al., 2013)“) add term „Table 1“. For examples („... in the literature (Okan et al., 2013) (Table 1)“).

Line 166-167: This sentence („The TPC, TFC and antioxidant ... given in Table 1“) must be deleted.

Line 213: Term “to be the “ replace with term “as“. Please, check and correct.

Line 233: Part in this sentence “ that could not be synthesized by human body” must be deleted.

Line 236: This part („In their study with pollen samples from the Bingöl region“) in sentence should be replace with („In bee pollen samples from the Bingöl region ...“).

Line 346-349: These sentences in method are confused and must be rewrite. Please, check and correct.

Author Response

Dear Reviewer,

General comments

Thank you for your constructive criticism. We understand your reserves. We tried to improve Manuscript (Introduction and phenolic profile section) and clarify the observed issues as much as it was possible. We hope that reviewer will understand some of our limitations that we had during this research.

Specific comments

All corrections were performed on the manuscript as you suggested. All changes are marked in color in the manuscript.

This manuscript is a resubmission of an earlier submission. The following is a list of the peer review reports and author responses from that submission.

Round 1

Reviewer 1 Report

Dear Authors, I think the topic of this article is very interesting. I suggest a revision of the English by a native speaker because there are some sentences that do not fit. The Latin names of plants (genus and species) should be written in italics.

In details:

Page 2 lines 28-30, 33-36, 41-42, 42-43, 48-49, 67-69, 73, 74 lack of bibliographic reference. Please add reference.

Page 2 line 38, the reference “Schmidt, 1996” is missing from the list of references at the bottom. Please add it. Please check that there are no other references missing from the list.

Page 2 line 45 please change the word 'humans' to 'beekeepers'.

Page 2 line 59 is preferred to what? Please clarify.

Page 3 lines 76-77, 77-79, 79-81, 81-83, 121-123, 123-124 lack of bibliographic reference. Please add reference.

Page 3 lines 104-116 The Latin names of plants (genus and species) should be written in italics. Please correct them.

Page 3 line 106 please change the word 'Apiacaee' to 'Apiaceae'.

Page 4 lines 125, 125-127, 127-129, 129-130, 130-131, 131-133, 133-135, 135-136, 136-137, 137-138, 138-139, 153-155, 155-156, lack of bibliographic reference. Please add reference.

Page 4 lines 158-163 for Italian bee pollen please refer to “Gabriele, M., Parri, E., Felicioli, A., Sagona, S., Pozzo, L., Biondi, C., Domenici, V., & Pucci, L. (2015). Phytochemical composition and antioxidant activity of Tuscan bee pollen of different botanic origins. Ital. J. Food Sci., 27:248:259.”

Page 5 line 183 Please change “16” to “sixteen”.

Page 5 line 187 “in different” instead of “in idifferent”.

Page 5 lines 199-202 lack of bibliographic reference. Please add reference.

Page 6 “antioxidant” capacity section. I would put this part together with the "total phenolic and flavonoid content" section because table 1 is discussed there and the reader has to go back and look for it. Otherwise split the table and put the part concerning this section here in a second table.

Page 6 lines 207-208, 208-209, 212-213, 213-214, 214, lack of bibliographic reference. Please add reference.

Page 6 line 213 what different methods? please elaborate.

Page 6 line 221 Anzer pollen has higher activity than what? Please clarify.

Page 7 lines 224-225 lack of bibliographic reference. Please add reference.

Page 7 “Fatty acid composition” section please refer to “Sagona, S., Pozzo, L., Peiretti, P. G., Biondi, C., Giusti, M., Gabriele, M., Pucci, L., & Felicioli, A. (2017). Palynological origin, chemical composition, lipid peroxidation and fatty acid profile of organic Tuscanian bee-pollen. Journal of Apicultural Research, 56(2), 136-143.”. Also these authors here discuss fatty acids as a function of human health.

Page 8 3.1 section. please add the magnification of the microscope used to identify the botanical species to which the pollen grains belong. Were a reference palynology and texts used? please add this part to the section.

Page 8 3.2 section. Please specify the percentage of ethanol used. Please specify ultrasonic bath conditions. please add centrifuge speed in rpm.

Page 9 lines 310 AlCl3, H2O, NaNO2 please put numbers as subscripts.

Page 9 lines 318, 319, 327 “10-2”, “10-3” please put -2 and -3 as superscript.

Page 9 lines 320, 327 please clarify what "x" stands for.

Page 9 lines 327, 328 Ce(SO4)2, H2O please put numbers as subscripts.

Page 9 3.7 section.  please add centrifuge speed in rpm. please specify what the extraction solution contains. please add at which frequency the sonicator was set.

Page 10. 3.8.2 please specify the concentration of NaCl used.

Page 10 line 369 please change “240 0C” and “260 0C” in “240 °C” and “260 °C”.

Page 10 conclusions. please specify that the pollen used was a mixture of pollen and had not been tested as individual botanical species.  

Reviewer 2 Report

  • This manuscript requires major English language copyediting.
  • The novelty of this study is unclear. It stated that no studies on the plant origin of the pollen samples from the Bayburt has been encountered. How is this goal related to the determination of phenolic compounds and fatty acids from bee pollen as indicated in the title?
  • The title itself does not represent the main goal of the paper as stated in the introduction.
  • How will other researchers benefit from this study? In particular, how would specific samples from a particular region in Turkey help future researchers in this study?
  • What is the main hypothesis of the study and why is there a need to conduct this kind of study?
  • I believe that to strengthen this manuscript, there should be more background about that specific region in Turkey where samples were taken from. What is so important about that region that warrants study and investigation for phenolic compounds and fatty acids from bee pollen?
  • How many samples were collected?
  • The study itself does not offer something novel. Yes, indeed, pollen samples may contain high antioxidant, etc but how does this translate and/or is related to that region in Turkey?
  • How does this manuscript differ from other previously conducted studies?
  • Please elaborate on the meaning of using this study for standardization purposes.
  • How does one ensure the homogeneity of the samples?